# Green Synthesis of Silver Nanoparticles with Extracts from *Kalanchoe fedtschenkoi*: Characterization and Bioactivities

**DOI:** 10.3390/biom14070782

**Published:** 2024-06-30

**Authors:** Jorge L. Mejía-Méndez, Gildardo Sánchez-Ante, Mónica Cerro-López, Yulianna Minutti-Calva, Diego E. Navarro-López, J. Daniel Lozada-Ramírez, Horacio Bach, Edgar R. López-Mena, Eugenio Sánchez-Arreola

**Affiliations:** 1Departamento de Ciencias Químico-Biológicas, Universidad de las Américas Puebla, Santa Catarina Mártir s/n, Cholula 72810, Puebla, Mexico; jorge.mejiamz@udlap.mx (J.L.M.-M.); monica.cerro@udlap.mx (M.C.-L.); yulianna.minuttica@udlap.mx (Y.M.-C.); jose.lozada@udlap.mx (J.D.L.-R.); 2Tecnologico de Monterrey, Escuela de Ingeniería y Ciencias, Av. Gral. Ramón Corona No 2514, Colonia Nuevo México, Zapopan 45121, Jalisco, Mexico; gildardo.sanchez@tec.mx (G.S.-A.); diegonl@tec.mx (D.E.N.-L.); 3Division of Infectious Diseases, Department of Medicine, University of British Columbia, Vancouver, BC V6H 3Z6, Canada

**Keywords:** nanotechnology, green synthesis, *Kalanchoe fedtschenkoi*, silver nanoparticles

## Abstract

In this work, the hexane, chloroform, and methanol extracts from *Kalanchoe fedtschenkoi* were utilized to green-synthesize silver nanoparticles (Kf_1_-, Kf_2_-, and Kf_3_-AgNPs). The Kf_1_-, Kf_2_-, and Kf_3_-AgNPs were characterized by spectroscopy and microscopy techniques. The antibacterial activity of AgNPs was studied against bacteria strains, utilizing the microdilution assay. The DPPH and H_2_O_2_ assays were considered to assess the antioxidant activity of AgNPs. The results revealed that Kf_1_-, Kf_2_-, and Kf_3_-AgNPs exhibit an average diameter of 39.9, 111, and 42 nm, respectively. The calculated ζ-potential of Kf_1_-, Kf_2_-, and Kf_3_-AgNPs were −20.5, −10.6, and −7.9 mV, respectively. The UV-vis analysis of the three samples demonstrated characteristic absorption bands within the range of 350–450 nm, which confirmed the formation of AgNPs. The FTIR analysis of AgNPs exhibited a series of bands from 3500 to 750 cm^−1^, related to the presence of extracts on their surfaces. SEM observations unveiled that Kf_1_- and Kf_2_-AgNPs adopted structural arrangements related to nano-popcorns and nanoflowers, whereas Kf_3_-AgNPs were spherical in shape. It was determined that treatment with Kf_1_-, Kf_2_-, and Kf_3_-AgNPs was demonstrated to inhibit the growth of *E. coli*, *S. aureus*, and *P. aeruginosa* in a dose-dependent manner (50–300 μg/mL). Within the same range, treatment with Kf_1_-, Kf_2_-, and Kf_3_-AgNPs decreased the generation of DPPH (IC_50_ 57.02–2.09 μg/mL) and H_2_O_2_ (IC_50_ 3.15–3.45 μg/mL) radicals. This study highlights the importance of using inorganic nanomaterials to improve the biological performance of plant extracts as an efficient nanotechnological approach.

## 1. Introduction

Over the last few decades, healthcare systems have been threatened by the increasing number of patients infected by pathogenic microorganisms [1] or diagnosed with oxidative stress-related disorders [2,3].

Infectious diseases constitute a broad category of illnesses caused by bacteria, fungi, viruses, or protozoa. The capacity of bacteria to resist the activity of antibiotics is known as antibacterial resistance (AR), and it is related to more than 2.8 million cases and 35,000 deaths per year in the United States [4]. Clinically, the treatment of bacterial pathogenic stains is based on the prescription of penicillin (e.g., amoxicillin and ampicillin), cephalosporins (e.g., cefazolin and ceftriaxone), or aminoglycosides (e.g., gentamicin and amikacin) [5,6,7]. Despite their possible efficacy, the administration of antibiotic regimens is problematic, due to the capacity of bacteria to develop a series of mechanisms to avoid their efficacy and, hence, elevate the rates of morbidity and mortality [8], as well as the associated healthcare costs [9].

Oxidative stress results from an imbalance between the levels of oxidants and antioxidants and can orchestrate the initiation and progression of many disorders, such as cardiovascular (e.g., hypertension and ischemic heart disease) [10], neoplastic (e.g., lung, breast cancer) [11,12], neurodegenerative (e.g., amyotrophic lateral sclerosis, Alzheimer’s disease, and Parkinson’s disease) [13,14], inflammatory (e.g., rheumatoid arthritis, asthma, and inflammatory bowel diseases) [15,16], and metabolic (e.g., non-alcoholic fatty liver disease, obesity, and dyslipidemia) diseases [17,18]. Depending on the diagnosed disease, the therapeutic regimens against oxidative stress involve the use of supplements (e.g., creatinine) [19], enzyme-targeted therapies (e.g., superoxide dismutase therapy) [20], lifestyle modifications (e.g., the adoption of different dietary patterns) [21], and pharmaceutical interventions [22].

Nanomedicine is an active research field wherein nanostructured matter is manipulated to treat, diagnose, monitor, or prevent diseases [23]. Noble metal-based nanoparticles (NPs) constitute a representative category of inorganic NMs that, due to their high surface area-to-volume ratio, modifiable features (i.e., size and shape), and superior drug transport capacity, are widely exploited to manufacture medical devices and fabricate platforms with superior antimicrobial, antioxidant, anticancer, and wound-healing properties [24,25].

Among the noble metals, silver constitutes a biologically active material that can be obtained from orthopedic materials, mining wastewater, or brazing alloys through bio-metallurgical and hydrometallurgical processes [26]. In nanomedicine, silver is utilized to manufacture biosensors [27], improve the antimicrobial activity of orthodontic adhesives [28], and elaborate the drug delivery systems of anticancer drugs such as doxorubicin [29]. As with other nanostructures, AgNPs can be prepared using both top-down (e.g., laser ablation and lithography) and bottom-up (e.g., chemical and solvothermal synthesis) techniques [30,31]. 

Green synthesis represents a cost-effective and environmentally friendly bottom-up technique wherein biological sources such as amino acids, proteins, enzymes, phytochemicals, or extracts from medicinal plants are utilized to promote the formation of AgNPs. In traditional medicine, plant-based extracts pose a major alternative with which to treat or ameliorate infections and neoplastic, inflammatory, neurological, or metabolic disorders. The therapeutic activities of plant-based extracts are attributed to their phytochemical composition, which is usually associated with the presence of alkaloids, flavonoids, carotenoids, polyphenols, saponins, and saccharides. Against multidrug-resistant bacteria, plant-based secondary metabolites are promising candidates for drug development since they can execute major effects by acting through various mechanisms; for example, they can disrupt their cell wall or membrane, form complexes with membrane lipids causing leakage, downregulate metabolic processes, or promote cell lysis [32]. 

In the same way, plant-based extracts represent a promising solution for the treatment of oxidative stress-related disorders since they can decrease lipid peroxidation markers, enhance the activity of antioxidant enzymes (e.g., catalase and superoxide dismutase), and reduce the generation of reactive oxygen species (ROS) [33]. Despite their great importance against AR and oxidative stress, the translation of plant-based extracts into clinical applications is hampered by their poor solubility and limited specificity. Green synthesis approaches can aid in overcoming current drawbacks related to the therapeutic application of plant-based extracts through the rational design of safe and bioactive nanostructures.

The family Crassulaceae includes a widely recognized group of medicinal plants that can exert multiple biological properties by means of the presence of bioactive secondary metabolites such as alkaloids, bufadienolides, polyphenols, and flavonoids [34]. The genus *Kalanchoe* belongs to this family, and it is widely recognized because of its ornamental, therapeutic, and nanomedical uses. In nanomedicine, polar extracts or isolated compounds from the genus *Kalanchoe* are widely utilized as reducing or capping agents to green-synthesize metal-based nanostructures. For example, extracts from *K. pinnata* have been used to prepare AgNPs with antibacterial and photocatalytic properties [35]. In contrast, extracts from *K. daigremontiana* have been employed to synthesize gold nanoparticles (AuNPs) with antioxidant and antiproliferative activities [36]. In addition, aqueous extracts from *K. fedtschenkoi* have been proposed to develop AuNPs with environmental applications [37]. 

*K. fedtschenkoi*, traditionally known as Vieira lavanda, is a poorly recognized and scientifically validated species from the genus *Kalanchoe*. Recently, our research group unveiled the phytoconstituents and biological activities of nonpolar and polar extracts from *K. fedtschenkoi* [38]. In that study, the presence of fatty acids, phytosterols, isoforms of vitamin E, and derivatives of quercetin or kaempferol in the hexane, chloroform, and methanol extracts from this species was demonstrated. In the same context, it was reported that treatment with these extracts exhibited strong cytotoxic and anti-inflammatory activities but showed poor performance against a panel of Gram-positive and Gram-negative bacteria strains.

Even though the synthesis of nanostructures from the extracts of *Kalanchoe* species constitutes an attractive alternative to enhance their biological performance by implementing cost-effective and sustainable experimental conditions, the use of extracts with distinct polarity from *K. fedtschenkoi* in green synthesis routes has been minimally evaluated. Here, we sought to continue expanding this research line by using the hexane, chloroform, and methanol extracts from *K. fedtschenkoi* to green-synthesize AgNPs through a simple experimental approach. The chemical composition of AgNPs was investigated using UV-Vis spectroscopy and Fourier transform infrared (FTIR) spectroscopy. Moreover, dynamic light scattering (DLS) analyses were employed to study the average particle size, size distribution, and zeta potential of AgNPs. The morphology of the formed AgNPs was confirmed with scanning electron microscopy (SEM), whereas the elemental distribution was assessed through energy-dispersive X-ray spectroscopy (EDS). The antibacterial activity of the obtained AgNPs was evaluated against *Escherichia coli*, *Pseudomonas aeruginosa*, and *Staphylococcus aureus*. The antioxidant activity of AgNPs was evaluated by utilizing the DPPH and H_2_O_2_ assays. This is the first study wherein extracts with different polarity from a *Kalanchoe* species are used to synthesize bioactive AgNPs.

## 2. Materials and Methods

### 2.1. Plant Material, Extract Obtention, and Preparation of AgNPs

The leaves of *K. fedtschenkoi* were obtained, identified, and macerated with hexane, chloroform, and methanol to obtain extracts as published in [38]. The formation of AgNPs using the hexane, chloroform, and methanol extracts from *K. fedtschenkoi* was carried out as reported in [35]. First, 1 mM of AgNO_3_ solution was prepared with Milli-Q water. Then, 50 mg of each extract was completely dissolved in DMSO of technical grade, and then they were incorporated into the original AgNO_3_ solution. The mixture was heated at 60 °C for 30 min under vigorous stirring in a dark place, centrifuged, and the resulting pellet was allowed to air-dry. The extracts were termed as Kf_1_, Kf_2_, and Kf_3_, whereas the AgNPs were designated as Kf_1_-, Kf_2_-, and Kf_3_-AgNPs, respectively.

### 2.2. Characterization of AgNPs

DLS analyses were performed using a Microtrac Nanotrac Wave II (Montgomeryville, PA, USA) instrument. The UV-vis spectra of the samples were recorded in 1-centimeter path-length rectangular quartz cuvettes from 200 to 800 nm on a Cary 60 spectrophotometer (Agilent Technologies, Santa Clara, CA, USA). The chemical composition of samples was studied using a Cary 630 FTIR spectrophotometer (Agilent Technologies, Santa Clara, CA, USA). The morphology of AgNPs was analyzed using a field emission high-resolution scanning electron microscope (Tescan MAIA 3, Brno-Kohoutovice/Czech Republic) equipped with a Bruker XFlash 6|30 EDS detector (Berlin, Germany); the operating energy was set at 10 keV. All readings were performed in triplicate.

### 2.3. Strains and Culture Media

The antibacterial activity of AgNPs was studied against Gram-positive and Gram-negative bacteria. Gram-positive bacteria comprised *Staphylococcus aureus* (ATCC 25923), whereas Gram-negative bacteria comprised *Escherichia coli* (ATCC 25922) and *Pseudomonas aeruginosa* (ATCC 14210) strains. Mueller–Hinton broth (B&D) was prepared to culture the tested strains at 37 °C under orbital shaking.

### 2.4. Microdilution Assay

Following previous protocols [39], the antibacterial activity of the extracts and AgNPs from *K. fedtschenkoi* was studied in a 96-well plate with a microdilution assay. Initially, the bacterial inocula were prepared to have a final optical density of 0.05 at 600 nm; then, they were dispensed into a 96-well plate together with 50, 100, 150, 200, 250, and 300 μg/mL of extracts and AgNPs from *K. fedtschenkoi*, with a final volume of 100 μL of medium per well. After this, the plate was placed under orbital shaking at 37 °C overnight. The next day, absorbances were determined at 600 nm in a Multiskan Skyhigh microplate spectrophotometer (Thermo Fisher Scientific, Waltham, MA, USA). Treatment with fosfomycin, vancomycin, or amikacin was considered a positive control. All experiments were performed in triplicate. The minimal inhibitory concentrations needed to inhibit the growth of 90% (MIC_90_) values were calculated based on the percentage of inhibition and the used concentrations in this study.

### 2.5. Antioxidant Assay

The DPPH assay was performed by dissolving 4 mg of DPPH (2,2-diphenyl-1-picrylhydrazyl) reagent into 100 mL of ethanol (technical grade) in a dark place for 2 h. After this, 200 μL of the resultant DPPH solution was mixed with 20 μL of extracts and AgNPs from *K. fedtschenkoi* in the following concentrations: 50, 100, 150, 200, 250, and 300 μg/mL. The resulting mixtures were kept in a dark place for 30 min, and absorbances were determined in a Cary 60 spectrophotometer (Agilent Technologies, Santa Clara, CA, USA) at 517 nm. The H_2_O_2_ scavenging activity was ascertained by mixing 70 μL at 40 mmol L^−1^ of H_2_O_2_ with 100 μL of extracts and AgNPs from *K. fedtschenkoi* at 50, 100, 150, 200, 250, and 300 μg/mL. The same spectrophotometer was utilized, but the wavelength was adjusted to 230 nm.

### 2.6. Statistical Analysis

Statistical differences were determined by a two-way analysis of variance (ANOVA), followed by Tukey’s mean separation test. Data were analyzed using OriginPro 2023 processing software 10.0.0.154 (OriginLab, Northampton, MA, USA).

## 3. Results

### 3.1. Synthesis and Formation of Kf_1_-, Kf_2_-, and Kf_3_-AgNPs

The formation of green-synthesized AgNPs arises from nucleation and growth phenomena. During these events, silver atoms aggregate and form small nuclei, which are the starting points necessary for the growth of AgNPs [40]. If applicable, the formed nuclei can adsorb extracts or isolated compounds and, hence, yield capped AgNPs. The presence of extracts as capping agents is desirable since they can provide stability and expand the biological performance of AgNPs [41]. This process is schematized in Figure 1A,B.

As illustrated in Figure 1C, the formation of Kf_1_-, Kf_2_-, and Kf_3_-AgNPs using non-polar and polar extracts from *K. fedtschenkoi* was monitored for 60 min. During this period, an evident color change from light yellow to bright brown was recorded, which can be attributed to the surface plasmon resonance (SPR) effect and confirms the successful formation of AgNPs. Similar results have been observed in studies where medicinal plants such as *Solanum xanthocarpum* or *Croton macrostachyus* have been considered for the green synthesis of AgNPs [42,43].

### 3.2. Characterization of Kf_1_-, Kf_2_-, and Kf_3_-AgNPs

#### 3.2.1. UV-Vis Spectroscopy Analysis

As depicted in Figure 2A, Kf_1_-, Kf_2_-, and Kf_3_-AgNPs exhibit various absorption peaks from 350 to 460 nm. The Kf_1_-AgNPs present two broad bands located at 359 and 420 nm, respectively. Comparably, Kf_2_-AgNP shows a broad peak at 356 nm and a sharp peak at 421 nm. Similar bands are visible for Kf_3_-AgNPs as two wide peaks appear at 351 and 434 nm, respectively. The identification of functional groups in Kf_1_-, Kf_2_-, and Kf_3_-AgNPs was performed by FTIR spectroscopy.

#### 3.2.2. FTIR Spectroscopy Analysis

As depicted in Figure 2B, the FTIR analysis of Kf_1_-, Kf_2_-, and Kf_3_-AgNPs exhibited a series of peaks from functional groups associated with the phytochemical content of extracts. For instance, the bands recorded at 3300 cm^−1^ may correspond to the stretching of the O-H and suggest the presence of phenolic compounds. The symmetrical and asymmetrical stretching of C-H bonds is exhibited within the region of 2915 to 2850 cm^−1^, whereas the stretching of C=O or C=C bonds is located from 1600 to 1150 cm^−1^. The bands displayed in these wavenumbers may be related to the existence of ketones, carboxylic acids, or esters and have previously been identified in the obtained extracts [38]. However, the small shifts identified during this analysis can be correlated to the capacity of Kf_1_-, Kf_2_-, and Kf_3_-AgNPs to act as capping agents.

#### 3.2.3. DLS Analysis

The average size and ζ-potential of the synthesized AgNPs are depicted in Figure 3A. In this figure, it can be seen that the size distribution of Kf_1_-AgNPs ranges from 102 to 33 nm, whereas the size distribution of Kf_2_-AgNPs ranges from 172 to 86 nm. The average size of the former is 39.9 nm, while the average size of the latter is 111 nm. Conversely, the size distribution of Kf_3_-NPs is found to be in the range of 102.2 to 36.1 nm. The calculated average size of Kf_3_-NPs is 42 nm.

As represented in Figure 3B, the calculated ζ-potential values of Kf_1_-, Kf_2_-, and Kf_3_-AgNPs by DLS were −20.5, −10.6, and −7.9 mV, respectively. In other studies, it has been documented that AgNPs synthesized with the aqueous extract of *K. pinnata* exhibited a ζ-potential value of −26.7 mV. In another report, it was revealed that the ζ-potential of AuNPs prepared with the ethanol extract from *K. daigremontiana* was found to be within the range of −20 mV [44]. 

The PDI constitutes the dispersion of the size populations of NPs in a medium, and it is further represented by numerical values that range from 0 to 0.1 [45]. Within this range, low PDI values are related to homogenous populations of NPs (<0.1) [46], whereas broader PDI values (>0.1) are associated with heterogeneous distributions of NPs [47]. According to these studies’ supplementary materials, the PDI of Kf_1_-, Kf_2_-, and Kf_3_-AgNPs were 0.04, 0.08, and 1.42, respectively.

#### 3.2.4. SEM and EDS Analysis of Kf_1_-, Kf_2_-, and Kf_3_-AgNPs

As depicted in Figure 4A, these NPs exhibit morphologies related to nano-popcorns [48]. In the same figure, an evaluation by EDS confirmed the presence of silver and carbon on their surfaces. The presence of the latter suggests the capacity of Kf_1_ extract to act as a capping agent.

In Figure 4B, it can be seen that Kf_2_-AgNPs possess nanoflower-like morphologies, which are structural arrangements that have also been recorded in studies on the use of extracts from *K. daigremontiana* [49]. According to EDS analysis, the presence of carbon also suggests the capacity of the Kf2 extract to act as a capping agent in Kf_2_-AgNPs. In contrast, the synthesis of Kf_3_-NPs exhibited the formation of spherical structures, wherein Kf_3_ extract also acted as a capping agent; this is presented in Figure 4C. The EDS spectrum of the samples is presented in Figure 4D, where it can be observed that Kf_1_-, Kf_2_-, and Kf_3_-AgNPs exhibited peaks of silver, oxygen, and carbon that correspond to the raw materials used during their synthesis or the compounds contained in extracts. The same figure also shows the presence of copper, which can be attributed to the materials utilized during the experimental process.

### 3.3. Antibacterial Activity of Kf_1_-, Kf_2_-, and Kf_3_-AgNPs

As depicted in Figure 5A, treatment with 50 and 100 μg/mL of Kf_1_ caused the death of 8.93 and 9.29% of *E. coli* cells, whereas treatment at 23.46, 25.17, and 26.35 μg/mL induced the death of 23.46, 25.17, and 26.35% of cells, respectively. Treatment with the same extract at 300 μg/mL resulted in the death of 27.25% of cells. Comparably, treatment with 50, 100, and 150 μg/mL of Kf_2_ resulted in the death of 5.70, 8.08, and 8.50% of cells of *E. coli*. The highest antibacterial activity of Kf_2_ against *E. coli* was determined at 200, 250, and 300 μg/mL, which resulted in the death of 8.91, 12.39, and 16.22% of cells. In the case of treatment with Kf_3_, it was noted that treatment with 50, 100, and 150 μg/mL caused the death of 10.16, 11.71, and 22.81% of cells, respectively. At 200, 250, and 300 μg/mL, treatment with Kf_3_ promoted the death of 23.89, 24.78, and 25.60% of cells.

When Kf_1_ was used to synthesize AgNPs, it was recorded that treatment with 50, 100, and 150 μg/mL resulted in the death of 66.56, 69.32, and 69.98% of *E. coli* cells, whereas at 200, 250, and 300 μg/mL, treatment with Kf_1_-AgNPs caused the death of 70.18, 71.93, and 72.12% of cells. This effect varied when Kf_2_- and Kf_3_-AgNPs were utilized against the same strain. For instance, it was determined that 50, 100, and 150 μg/mL of the former promoted the death of 47.74, 49.87, and 50.01% of cells. The highest antibacterial effect of Kf_2_-AgNPs was observed at 200, 250, and 300 μg/mL since it resulted in the death of 50.96, 53.12, and 65.06% of cells, respectively. In the case of the latter, it was determined that 28.64, 34.92, and 36.38% of cells died during treatment with 50, 100, and 150 μg/mL of Kf_3_-AgNPs. At 200, 250, and 300 μg/mL, Kf_3_-AgNPs caused the death of 37.75, 39.68, and 40.20% of *E. coli* cells. These results are presented in Figure 5B.

Against *P. aeruginosa*, treatment with 50, 100, and 150 μg/mL of Kf_1_ induced the death of 2.81, 8.74, and 8.80% of cells, whereas treatment at 200, 250, and 300 μg/mL resulted in the death of 10.18, 12.99, and 13.36% of cells. Similarly, treatment with 50, 100, and 150 μg/mL of Kf_2_ caused the death of 2.24, 9.68, and 10.24% of *P. aeruginosa* cells. Treatment with Kf_2_ at 200, 250, and 300 μg/mL resulted in the death of 11.55, 12.05, and 12.24% of cells. Among the tested extracts, Kf_3_ exerted the highest antibacterial activity toward *P. aeruginosa*. For example, 50, 100, and 150 μg/mL of Kf_3_ induced the death of 18.05, 36.91, and 83.76% of cells, whereas 200, 250, and 300 μg/mL promoted the death of 84.51, 87.44, and 90.06% of cells. These findings are illustrated in Figure 5C.

In comparison with these findings, treatment with Kf_1_-AgNPs resulted in the death of 2.81, 8.74, and 8.80% of *P. aeruginosa* cells at 50, 100, and 150 μg/mL, respectively. Treatment with 200, 250, and 300 μg/mL of Kf_1_-AgNPs resulted in the death of 10.18, 12.99, and 13.36% of cells. When Kf_2_-AgNPs were used against *P. aeruginosa*, the death of 2.42, 9.68, and 10.24% of cells was determined at 50, 100, and 150 μg/mL, respectively. At 200, 250, and 300 μg/mL, treatment with Kf_2_-AgNPs caused the death of 11.55, 12.05, and 12.42% of cells. Regarding the activity of Kf_3_-AgNPs toward *P. aeruginosa*, it was noted that treatment at 50, 100, and 150 μg/mL resulted in the death of 18.05, 36.91, and 83.76% of cells, respectively. At 200, 250, and 300 μg/mL, Kf_3_-AgNPs induced the death of 84.50, 87.44, and 90.06% of cells (see Figure 5D).

As represented in Figure 5E, no significant activity against *S. aureus* during treatment with Kf_1_ was observed at 50, 100, or 150 μg/mL. However, it caused the death of 38.62, 65.30, and 66.99% cells at 200, 250, and 300 μg/mL, respectively. Conversely, treatment with 50, 100, and 150 μg/mL resulted in the death of 21.20, 25.42, and 31.60% *S. aureus* cells, whereas at 200, 250, and 300 μg/mL, treatment resulted in the death of 38.90, 62.50, and 66.85% of cells. Toward the same strains, treatment with Kf_3_ exhibited the highest antibacterial activity since it was recorded that at 50, 100, and 150 μg/mL, it resulted in the death of 22.47, 50.98, and 62.21% of cells, respectively. At 200, 250, and 300 μg/mL, treatment with Kf_3_ induced death in 68.39, 69.80, and 73.17% of cells.

Conversely, treatment with 50 and 100 μg/mL of Kf_1_-AgNPs caused the death of 49.57 and 50.56% of *S. aureus* cells, respectively. Interestingly, it was determined that treatment with 150 or 200 μg/mL caused the death of 67.27% of cells. At 250 and 300 μg/mL, treatment with Kf_1_-AgNPs induced the death of 79.21 and 94.66% of *S. aureus* cells. During treatment with Kf_2_-AgNPs, it was observed that 50, 100, and 150 μg/mL caused the death of 2.00, 8.00, and 31.32% of *S. aureus* cells. At 200, 250, and 300 μg/mL, it resulted in the death of 36.23, 37.92, and 53.37% of cells. The highest antibacterial activity was observed during treatment with Kf_3_-AgNPs as it caused the death of 78.51, 78.65, and 79.07% of *S. aureus* cells at 50, 100, and 150 μg/mL, whereas, at 200, 250, and 300 μg/mL, it induced the death of 79.35, 80.74, and 81.32% of *S. aureus* cells (see Figure 5F). The results generated during the antibacterial assay demonstrated that the green synthesis of AgNPs represents a viable approach to enhancing the antibacterial activity of extracts from *K. fedtschenkoi* against *E. coli*, *P. aeruginosa*, and *S. aureus*. The calculated MIC_90_ of samples against the tested bacteria strains are presented in Table 1.

### 3.4. Antioxidant Activity of Kf_1_-, Kf_2_-, and Kf_3_-AgNPs

Here, the antioxidant activity of extracts from *K. fedtschenkoi* and the synthesized AgNPs was analyzed by the DPPH and H_2_O_2_ assays, respectively. As illustrated in Figure 6A, the capacity of extracts from *K. fedtschenkoi* to scavenge DPPH radicals increased in a dose-dependent manner. For example, treatment with 50, 100, and 150 μg/mL of Kf_1_ inhibited the formation of 12.85, 24.87, and 44.26% of DPPH radicals, respectively. At 200, 250, and 300 μg/mL, treatment with Kf_1_ scavenged 75.09, 75.83, and 80.37% of radicals, respectively. In contrast, treatment with 50 and 100 μg/mL of Kf_2_ resulted in 15.73 and 17.23% of scavenged DPPH radicals, respectively.

Moreover, treatment with 150, 200, and 250 μg/mL of Kf_2_ resulted in the inhibition of 64.17, 78.14, and 79.49% of radicals, respectively. For this extract, the highest antioxidant activity was recorded at 300 μg/mL as it caused the inhibition of 85.14% of DPPH radicals. In contrast to the Kf_1_ and Kf_2_ extracts, Kf_3_ demonstrated superior scavenging activity at 50, 100, and 150 μg/mL, where 71.18, 75.04, and 75.55% of inhibited radicals were determined, respectively. At higher concentrations, treatment with 200, 250, and 300 μg/mL of Kf_3_ scavenged 77.62, 81.41, and 84.66% of DPPH radicals, respectively.

The DPPH scavenging activity of Kf_1_-, Kf_2_-, and Kf_3_-AgNPs is presented in Figure 6B. As observed, treatment with 50, 100, and 150 μg/mL of Kf_1_-AgNPs scavenged 53.59, 54.29, and 54.46% of radicals, respectively. At 200, 250, and 300 μg/mL, treatment with Kf_1_-AgNPs inhibited the generation of 92.07, 93.20, and 93.81% of DPPH radicals, respectively. In the case of Kf_2_-AgNPs, it can be seen that treatment with 50 and 100 μg/mL scavenged 44.21 and 50.23% of free radicals, whereas at 150 and 200 μg/mL, it inhibited the formation of 52.69 and 55.07% of DPPH radicals, respectively. At 250 and 300 μg/mL, Kf_2_-AgNPs scavenged 55.49 and 77.31% of free radicals, respectively. For Kf_3_-AgNPs, it was recorded that 77.80, 81.17, and 81.76% of DPPH radicals were inhibited at 50, 100, and 150 μg/mL, respectively. At 200, 250, and 300 μg/mL, Kf_3_-AgNPs scavenged 82.03, 83.12, and 85.75% of DPPH radicals, respectively. 

The DPPH scavenging activity of extracts from *K. fedtschenkoi* and the synthesized AgNPs was compared with Qu, which, at 50, 100, and 150 μg/mL, inhibited the generation of 77.80, 81.17, and 81.76% of free radicals, respectively. It was registered that 200, 250, and 300 μg/mL of Qu scavenged 82.03, 83.12, and 85.75% of DPPH radicals, respectively. According to the results obtained during the DPPH assay, the IC_50_ values were calculated for extracts from *K. fedtschenkoi* and AgNPs and are compiled in Table 2, which also contains the determined IC_50_ values of samples against H_2_O_2_ radicals.

The capacity of extracts and AgNPs from *K. fedtschenkoi* to inhibit the formation of H_2_O_2_ radicals is depicted in Figure 6C,D. As noted in Figure 6C, treatment with Kf_1_ at 50, 100, and 150 μg/mL scavenged 20.94, 28.43, and 35.76% of H_2_O_2_ radicals, respectively. At 200, 250, and 300 μg/mL, treatment with Kf_1_ scavenged 51.89, 55.62, and 77.85% of H_2_O_2_ radicals, respectively. It can be observed in the same figure that treatment with Kf_2_ inhibited the generation of 56.13, 73.60, and 82.23% of H_2_O_2_ radicals, respectively. Interestingly, treatment with 200 μg/mL of Kf_2_ resulted in the scavenging of 83.53% of H_2_O_2_ radicals, whereas treatment with 250 and 300 μg/mL inhibited the formation of 100% of radicals. For extracts, the highest scavenging activity toward H_2_O_2_ radicals was achieved with Kf_3_, as it scavenged 74.02, 74.49, and 74.67% at 50, 100, and 150 μg/mL, respectively. At 200, 250, and 300 μg/mL, treatment with Kf_3_ resulted in the inhibition of 75.09, 75.27, and 75.49% of H_2_O_2_ radicals, respectively. The activity of extracts from *K. fedtschenkoi* in this assay was enhanced using AgNPs.

As represented in Figure 6D, treatment with Kf_1_-AgNPs scavenged 89.52, 81.97, and 92.68% of H_2_O_2_ radicals at 50, 100, and 150 μg/mL, respectively. At 200, 250, and 300 μg/mL, treatment with Kf_1_-AgNPs inhibited the generation of 94.72, 95.46, and 96.60% of radicals, respectively. Similarly, treatment with 50, 100, and 150 μg/mL of Kf_2_-AgNPs resulted in the scavenging of 84.96, 86.04, and 86.14% of H_2_O_2_ radicals, respectively. At 200, 250, and 300 μg/mL, treatment with Kf_2_-AgNPs scavenged 94.89, 94.93, and 95.26% of radicals, respectively. In contrast to these results, treatment with 50, 100, and 150 μg/mL of Kf_3_-AgNPs resulted in the inhibition of 30.29, 69.70, and 70.80% of H_2_O_2_ radicals, respectively. Treatment with 200, 250, and 300 μg/mL of Kf_3_-AgNPs resulted in the scavenging of 75.09, 75.27, and 75.49% of H_2_O_2_ radicals. The calculated IC_50_ values of samples against H_2_O_2_ radicals are presented in Table 2.

## 4. Discussion

The chemical diversity of the secondary metabolites contained in extracts from medicinal plants has been advantageous for drug development, formulating drugs against different disorders. In the clinical pipeline, the administration of extracts is complicated, due to their poor solubility, bioavailability, and limited specificity. Over the last few decades, green nanotechnology has been exploited as an environmentally friendly and sustainable approach to overcoming the current drawbacks associated with the application of medicinal plants. Extracts from medicinal plants can promote the formation of bioactive metal-based nanostructures by means of the presence of biomolecules with the capacity to act as reducing or capping agents under controlled experimental conditions. 

As previously mentioned, our research group recently demonstrated that hexane (Kf_1_), chloroform (Kf_2_), and methanol (Kf_3_) extracts from *K. fedtschenkoi*, obtained by maceration and with distinct extraction yields (0.07, 0.05, and 0.08%), exerted strong cytotoxic effects against THP-1 cells within the range of 50–200 μg/mL, and also significantly promoted the secretion of pro-inflammatory cytokines such as IL-10 at 50 μg/mL. In the case of Kf_2_ and Kf_3_ extracts, it was reported that treatment at 150 μg/mL (MIC value) inhibited the growth of *P. aeruginosa* and methicillin-resistant *Staphylococcus aureus* (MRSA), respectively. In that study, the observed activities were correlated with their phytochemical content, which in the case of Kf_1_ and Kf_2_ extracts was documented for the first time by GC/MS, whereas the Kf_3_ extract was assessed by HPLC analyses. The phytochemistry of Kf_1_ and Kf_2_ extracts comprise, predominantly, the isoforms of tocopherol (α, δ, and ε), and phytosterols such as stigmasterol and simiarenol. In contrast, Kf_3_ extract was demonstrated to be an abundant source of the glycosidic derivatives of quercetin and kaempferol, which are compounds with the capability to act, either alone or in synergy, as reducing agents during the synthesis of AgNPs.

As presented in Figure 1A, the synthesis of Kf_1_-, Kf_2_-, and Kf_3_-AgNPs was initially monitored through color changes from light yellow to bright brown. To confirm their adequate formation, this study integrated the use of different spectroscopy and microscopy techniques to unveil the physicochemical characteristics of Kf_1_-, Kf_2_-, and Kf_3_-AgNPs. UV-Vis spectroscopy is a widely implemented spectroscopy technique that is required to evaluate the size, shape, and formation of noble metal-based NPs [50]. In accordance with the obtained results, it was observed that Kf_1_-, Kf_2_-, and Kf_3_-AgNPs exhibit sharp and broad bands within the range of 350–600 nm, which are frequently reported during the green synthesis of nanostructures and can be correlated with their SPR effect [51]. In addition, it was noted that the UV-vis spectroscopy analyses of AgNPs exhibited the presence of additional wide bands, which may be associated with the existence of secondary metabolites from the extracts on their surfaces (see Figure 2A). Considering these results, the chemical composition of Kf_1_-, Kf_2_-, and Kf_3_-AgNPs was also investigated through FTIR spectroscopy.

FTIR spectroscopy is based on the absorption of infrared light by nucleic acids, proteins, carbohydrates, or metals. Each sample exhibits a distinctive spectrum fingerprint in this technique that can be recognized and differentiated from other molecules [52]. As noted in Figure 2B, the FTIR analysis of Kf_1_-, Kf_2_-, and Kf_3_-AgNPs demonstrated the characteristic peaks of secondary metabolites from *K. fedtschenkoi* extracts containing ketones, carboxylic acids, esters, and hydroxyl groups. Based on our previous studies, it was noted that the peaks exhibited a small shift within the considered wavenumber region (4000–400 cm^−1^), suggesting that the extracts can act as capping agents. In the same context, it can be seen from Figure 2B that AgNPs do not exhibit the peaks of the solvents (hexane, chloroform, and methanol) utilized to obtain the extracts, which may be related to the fact that solvents were carefully eliminated by rotary evaporation during extract preparation to avoid potential toxicity hazards. In the same context, the absence of peaks associated with the presence of solvents among the synthesized AgNPs may be due to the implemented air-drying process. 

DLS, also known as quasi-elastic light scattering or correlation spectroscopy, is a non-invasive technique that is used to evaluate the average size and ζ-potential of nanomaterials. Since nanomaterials are in constant Brownian motion when in suspension, DLS can also be used to determine their PDI. The size of the NPs influences their interaction with biological elements, their fate in organisms, and toxicity, and decides their performance, either in vitro or in vivo [53]. There are distinct approaches that have previously been followed to synthesize nanostructures with species from the genus *Kalanchoe*, specifically with polar extracts such as aqueous and ethanol extracts.

For example, in a recent study, an aqueous extract from the leaves of *K. pinnata* has been used to synthesize 37.8–43.8 nm AgNPs [35]. Similarly, the aqueous, ethanol, and isopropanol extracts from *K. daigremontiana* have been utilized to promote the formation of 14.3, 7.7, and 17.5 nm AuNPs, respectively [36]. The ethanol extract from the same species has been considered in the formation of 32 nm AgNPs [44], whereas the aqueous extract from *K. fedtschenkoi* has been utilized to fabricate 32.7 nm AuNPs [37]. Here, we envisioned the use of a green synthesis approach as an advantageous alternative to improve the biological activities of Kf_1_ and Kf_2_ extracts, which, in accordance with our previous study, performed poorly against pathogenic bacteria strains. In addition, their use for the green synthesis of AgNPs was expected to expand our knowledge of the importance of extracts from *Kalanchoe* species that are not necessarily polar. Conversely, the discrepancies between the provided evidence with the recorded sizes of Kf_1_-, Kf_2_-, and Kf_3_-AgNPs may be due to variabilities in the amount of extract used, the concentration of AgNO_3_, synthesis conditions such as temperature, and stirring, and the experimental setup.

The ζ-potential of Kf_1_-, Kf_2_-, and Kf_3_-AgNPs was also evaluated by DLS. The ζ-potential refers to the electric potential found in the interfacial double layer of a dispersed particle. For NPs, high positive or negative ζ-potential values (+30 or −30 mV) are correlated with their stability in dispersion, resistance to aggregation, and capacity to evade an increase in size [54]. In contrast to size analyses, the evaluation of ζ-potential from green-synthesized nanostructures with extracts from *Kalanchoe* species is not frequently considered. For instance, only one study demonstrated that AuNPs that were synthesized with the aqueous, ethanol, and isopropanol extracts from *K. fedtschenkoi* exhibited different ζ-potential values: −25.6, −34.9, and −42.3 mV, respectively [34]. In this case, the similarity regarding the negative charge of Kf_1_-, Kf_2_-, and Kf_3_-AgNPs with samples from the mentioned study can be attributed to the presence of related electronegative functional groups on their surfaces. 

Among electron microscopy techniques, SEM constitutes a powerful approach for analyzing the size, shape, and topography of micro- and nanostructures. In comparison with other electron-based methods, SEM can be coupled with additional techniques (e.g., transmission electron microscopy or EDS) that are required to expand our physicochemical knowledge of NMs. The morphology and elemental composition of the synthesized AgNPs were evaluated using SEM and EDS, respectively. The SEM analyses of Kf_1_-, Kf_2_-, and Kf_3_-AgNPs revealed that they adopted different structural arrangements, even though the same experimental protocol was implemented. Currently, there are no studies devoted specifically to the investigation of the influence of plant extract polarity and their phytochemical content on the morphological features of NMs. However, similar results have been observed with the use of extracts from *Kalanchoe* species. For example, it was determined that polar extracts from *K. daigremontiana* favored the yield of nanoflowers with representative nano-petals and defined spectroscopy behavior [49], whereas extracts from *K. pinnata* promoted the formation of spherical AgNPs with photocatalytic and antibacterial activities [55]. The differences between the structural arrangement of Kf_1_-, Kf_2_-, and Kf_3_-AgNPs can be attributed to interactions among the phytoconstituents of extracts with metallic ions during the synthesis process. However, the discrepancies between the determined sizes during DLS and SEM analyses occurred because the samples were not filtered or sonicated prior to SEM evaluation. This was performed with the purpose of observing the natural structural arrangement of Kf_1_-, Kf_2_-, and Kf_3_-AgNPs. In the same context, variabilities among the recorded sizes of AgNPs from *K. fedtschenkoi* may be associated with the experimental process for analyzing samples by DLS, whereby they were filtered to remove insoluble extracts, residuals, and agglomerates.

Current innovative approaches to combat AR encompass the rational design and use of phage therapies, antimicrobial peptides, host-directed therapies, microbiome modulation, antivirulence strategies, repurposing existing drugs, and NMs [56,57]. Here, fosfomycin, amikacin, and vancomycin were utilized as positive controls to compare the antibacterial activity of extracts or AgNPs from *K. fedtschenkoi*, due to reported bacterial sensibility. In contrast to other Gram-negative bacteria, *E. coli* is a rod-shaped strain and is a member of the Enterobacteriaceae family; it is frequently localized in the lower intestinal tract of humans and constitutes part of the normal gut microbiota. However, common sources of *E. coli* include water sources, soil, and animal-origin food products [58,59]. In healthcare systems, the prevalence of *E. coli* is problematic since it possesses the capacity to exhibit a series of intrinsic resistance mechanisms and causes infections that are challenging to treat among immunocompromised patients and the elderly [60]. In the same context, *P. aeruginosa* is a rod-shaped strain belonging to the Pseudomonadaceae family that is characterized by its multidrug-resistant nature, which emphasizes the need for improved infection control and antibacterial stewardship programs. It also has the capacity to contribute to the development of nosocomial infections [61]. Frequent sources of *P. aeruginosa* are medical equipment, water systems, and contaminated surfaces [62].

Conversely, *S. aureus* is a Gram-positive spherically shaped strain that belongs to the Staphylococcaceae family; it is responsible for causing life-threatening systemic diseases such as endocarditis and sepsis due to its widespread production of virulence factors and surface proteins [63] and its enhanced ability to colonize and persist in healthcare and community settings [64]. Due to its high colonizing capacity, the sources of *S. aureus* include infected individuals, medical equipment, contaminated surfaces, and livestock such as cattle and pigs [65]. In contrast to other alternatives, the use of green-synthesized AgNPs is advantageous since they can exert multifunctional features and inhibit the growth of pathogenic bacteria by disrupting the integrity of their cell wall, inducing oxidative stress, interfering with nutrient uptake and enzyme activity, and downregulating their capacity to release virulence factors [66]. These mechanisms are illustrated in Figure 7.

In contrast to amikacin and vancomycin, fosfomycin is a broad-spectrum phosphonic-type antibiotic that inhibits bacterial cell wall synthesis. In clinical practice, it is frequently prescribed to treat urinary tract infections, which are predominantly caused by *E. coli*. When administered alone against other Gram-negative bacteria such as *P. aeruginosa*, treatment with fosfomycin is ineffective, due to AR mechanisms [67]. Amikacin is an aminoglycoside antibiotic that is effective for treating infections caused by Gram-negative multidrug-resistant strains such as *P. aeruginosa* in immunocompromised patients [68]. Against *E. coli*, other antibiotic-based regimens are considered instead because of its susceptibility to and lower capacity for resistance against treatment. Vancomycin is a glycopeptide antibiotic administered against *S. aureus* or MRSA infections [69]. Against *E. coli* or *P. aeruginosa*, vancomycin is not prescribed since these bacteria possess more complex and less permeable outer membranes that lack the specific target sites of this antibiotic.

In accordance with other metal-based NMs, the antibacterial activity of Kf_1_-, Kf_2_-, and Kf_3_-AgNPs may be associated with their physicochemical properties, especially with their shape, size, and negative charge. In the case of Kf_1_- and Kf_2_-AgNPs, which exhibited a nano-popcorn or nanoflower-like structure, it has been reported that their sharp edges and high surface area-to-volume yield multiple interactions with bacterial cells; hence, they cause the disruption of the bacterial cell membrane and cause the enhanced release of metal ions, resulting in aberrations in bacterial metabolism [70]. 

Comparably, the antibacterial properties of Kf_3_-AgNPs can be related to their spherical arrangement, which favors their interaction with bacterial cell components and their capacity to upregulate the production of ROS [71]. The size of green-synthesized NMs is a major parameter that significantly influences their antibacterial activity; the average size of Kf_1_-, Kf_2_-, and Kf_3_-AgNPs was 39.9, 111, and 42 nm, respectively. Recent scientific evidence suggests that small-sized NMs (<100 nm) tend to exhibit a high surface area-to-volume ratio that enhances their interaction with bacterial cells. Additionally, their small size aids in their internalization and localization among cellular organelles, leading to bacterial cell death [72]. 

Unlike large NMs (>100 nm), small-sized NMs are considered more effective antibacterial agents since they are able to bypass or overcome the resistance mechanisms of bacterial strains. The negative surface of green-synthesized NMs can strongly dictate their performance toward multi-drug resistant bacteria; the calculated ζ-potential values of Kf_1_-, Kf_2_-, and Kf_3_-AgNPs were −25.6, −34.9, and −42.3 mV, respectively. In contrast to positively charged NMs, green synthesized NMs with negative ζ-potential values present enhanced electrostatic interaction with positively charged bacterial cell components, which can result in membrane thinning, pore formation, the disruption of membrane structure, leakage of cellular content, or the inhibition of biofilm formation [73]. 

Here, it was recorded that treatment with either extracts or Kf_1_-, Kf_2_-, and Kf_3_-AgNPs decreased the viability of Gram-positive and Gram-negative bacteria during treatment with concentrations ranging from 50 to 300 μg/mL. However, no statistical differences were determined in comparison to treatment with fosfomycin, amikacin, or vancomycin. Even though the obtained results regarding the antibacterial activity of Kf_1_-, Kf_2_-, and Kf_3_-AgNPs suggest their potential use as antibacterial agents and establish the possibility of consolidating these innovative research fields, there is a need to improve their antibacterial activity by modifying the experimental design or integrating additional bioactive agents since, according to their MIC_90_ values, their antibacterial activity against *E. coli* and *S. aureus* is still low. Interestingly, it was noted that because of their MIC_90_ values, treatment with Kf_1_ (MIC_90_ 438.41 μg/mL), Kf_2_ (MIC_90_ 425.36 μg/mL), and Kf_3_ (MIC_90_ 254.27 μg/mL) extracts exhibited higher levels of activity than AgNPs against *P. aeruginosa*. This phenomenon may be involved with the synergistic effect of the phytochemicals contained in *K. fedtschenkoi* extracts and their capacity to target multiple cellular mechanisms during treatment.

According to their origin, antioxidants are classified as either natural or synthetic. In comparison to natural oxidants (e.g., carotenoids, pomegranates, polyphenols, and vitamins) [74,75,76], synthetic antioxidants (e.g., oxides or metallic NPs) are preferred due to their higher stability, wide-ranging biological performance, and ease of production [77]. As with other metallic NPs, green-synthesized AgNPs can decrease the generation of free radicals through two main mechanisms (chain-breaking and prevention), and there are distinct approaches by which the antioxidant activity of green-synthesized AgNPs can be determined; for instance, the DPPH [78], ABTS [79], or FRAP [80] assays. Together with these techniques, the antioxidant activity of green-synthesized AgNPs can also be explored by considering the free radicals involved in pathological disorders; these include H_2_O_2_ [81], OH [82], and ^1^O_2_ radicals [83].

In contrast to other methods, the DPPH assay is a simple, reliable, and sensible technique to evaluate the antioxidant activity of a variety of NMs; for example, polymeric systems such as chitosan NPs, combined with natural polymers (i.e., alginate, pectin, and β-cyclodextrin) [84], lanthanide-based NPs such as CeO_2_-NPs, synthesized with *Mentha royleana* leaf extract [85], and metal-oxide nanostructures such as ZnO-NPs prepared with *Scoparia dulcis* plant extract [86]. In the same context, the H_2_O_2_ assay is a versatile and reproducible approach that provides a useful insight into the antioxidant activity of synthetic antioxidants and its relevance to oxidative stress.

Based on their IC_50_ values, Kf_1_ and Kf_2_ extracts can be considered moderate antioxidants for DPPH radicals, given their IC_50_ values of 164.54 and 153.64 μg/mL. In contrast, Kf_3_ extract can be considered a strong antioxidant due to its IC_50_ value: 1.92 μg/mL. Similarly, Kf_1_- and Kf_2_-AgNPs exerted moderate antioxidant activity, and Kf_3_-AgNPs exhibited high antioxidant performance. The calculated IC_50_ values for these samples were 57.02, 119.38, and 2.09 μg/mL, respectively. Tests against H_2_O_2_ indicate that the antioxidant activity of Kf_1_ extract (IC_50_ 197.52 μg/mL) is moderate, whereas Kf_2_ (IC_50_ 1.02 μg/mL) and Kf_3_ (IC_50_ 3.54 μg/mL) display strong antioxidant capacity. On the one hand, the capacity of AgNPs or extracts from *K. fedtschenkoi* to scavenge DPPH radicals arises from their ability to donate hydrogen atoms or electrons to aid in the reduction of DPPH into its stable form. On the other hand, treatment with the synthesized AgNPs can inhibit the formation of H_2_O_2_ radicals by means of the capability of their surface to act as a catalyst, facilitating electron transfer and, hence, the reduction of H_2_O_2_ into water and oxygen (see Figure 7). The results obtained during the antioxidant evaluation of the extracts and AgNPs from *K. fedtschenkoi* revealed their capacity to disrupt the generation of free radicals and suggest their possible application in oxidative stress-related disorders.

## 5. Conclusions

The green synthesis of bioactive nanostructures has become a highly active research field in recent years. This study demonstrated for the first time the use of non-polar and polar extracts from *K. fedtschenkoi* to green-synthesize AgNPs. In agreement with the obtained results, it was observed that certain experimental conditions (temperature and stirring) favored the obtention of AgNPs with different structural arrangements, specifically nano-popcorns, nanoflowers, and spheres. The characterization of the fabricated NMs by DLS, UV-VIS, and FTIR confirmed their formation and revealed that they possess a defined size distribution, surface charge, PDI, and functional groups that are possibly attached to the surfaces of AgNPs. In contrast, analyses by EDS unveiled the distribution of carbon on the surface of AgNPs, suggesting that extracts from *K. fedtschenkoi* can act as capping agents and provide stability and multifunctional features such as antibacterial and antioxidant activities. Against bacteria, it was noted that treatment with extracts can decrease the viability of *E. coli, P. aeruginosa,* and *S. aureus* within the range of 50–300 μg/mL. Even though no statistical differences were recorded, it was noted that treatment at initial concentrations of Kf_1_-, Kf_2_-, or Kf_3_-AgNPs can compromise the growth of higher proportions of the cultured Gram-positive and Gram-negative strains. Against DPPH and H_2_O_2_ radicals, it was determined that extracts and AgNPs could act as high or moderate antioxidants according to the calculated IC_50_ values. The findings provided by this work expand the potential of inorganic NMs to improve the therapeutic performance of extracts from medicinal plants and continue assessing the importance of considering innovative alternatives to treat bacterial infections or oxidative stress-related diseases. The size, shape, surface charge, and capping agents of the developed AgNPs converted them into suitable agents with potential applications in controlling the growth of food-borne pathogens through smart packaging development, in the design of antimicrobial coatings for medical devices, the production of water purification systems, the formulation of disinfectants, and crop growth protection. 

## Figures and Tables

**Figure 1 biomolecules-14-00782-f001:**
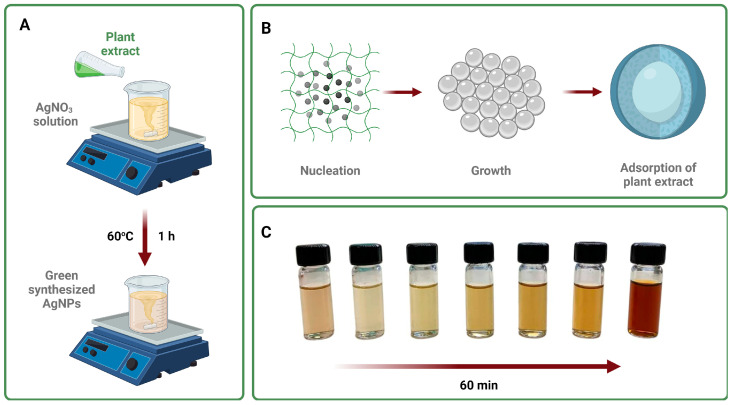
(**A**) General representation of the experimental design required for the green synthesis of AgNPs utilizing plant extracts, (**B**) the mechanism of formation of green synthesis of AgNPs, and (**C**) evaluation of the color change of Kf_1_-, Kf_2_-, and Kf_3_-AgNPs over 0, 10, 20, 30, 40, 50, and 60 min.

**Figure 2 biomolecules-14-00782-f002:**
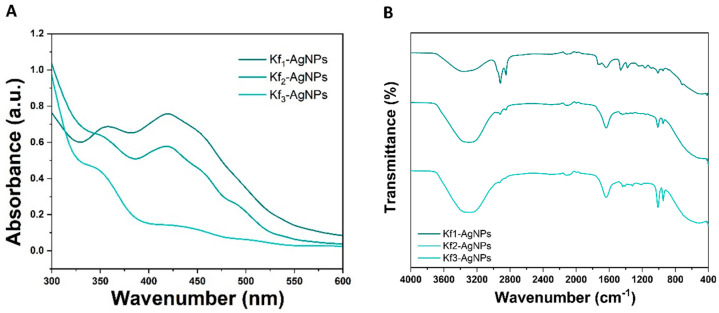
(**A**) UV-Vis spectroscopy and (**B**) FTIR analyses of Kf_1_-, Kf_2_-, and Kf_3_-AgNPs.

**Figure 3 biomolecules-14-00782-f003:**
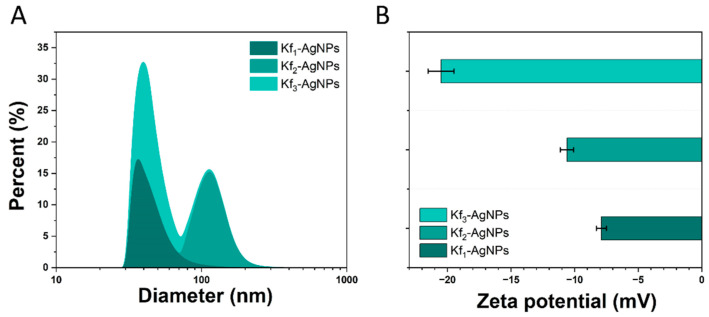
(**A**) Size distribution and (**B**) ζ-potential of Kf_1_-, Kf_2_-, and Kf_3_-AgNPs.

**Figure 4 biomolecules-14-00782-f004:**
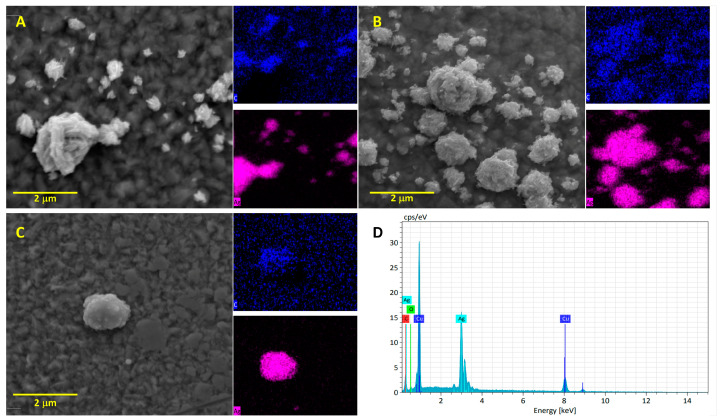
SEM analysis of (**A**) Kf_1_-, (**B**) Kf_2_-, (**C**) Kf_3_-AgNPs, and (**D**) EDS analysis. Insets in blue or pink are related to the distribution of carbon and silver among the tested samples.

**Figure 5 biomolecules-14-00782-f005:**
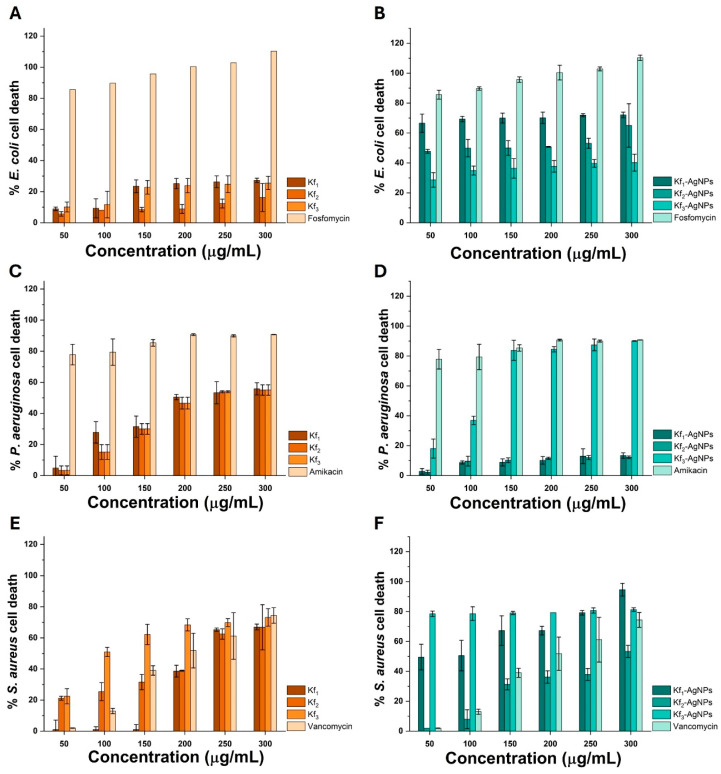
Antibacterial activity of extracts (Kf_1_, Kf_2_, and Kf_3_) and AgNPs (Kf_1_-, Kf_2_-, and Kf_3_-AgNPs) drawn from *K. fedtschenkoi* against (**A**,**B**) *E. coli*, (**C**,**D**) *P. aeruginosa*, and (**E**,**F**) *S. aureus*. Based on their sensibility, positive controls for each strain included fosfomycin, amikacin, and vancomycin. The mean ± SD of three independent experiments is shown.

**Figure 6 biomolecules-14-00782-f006:**
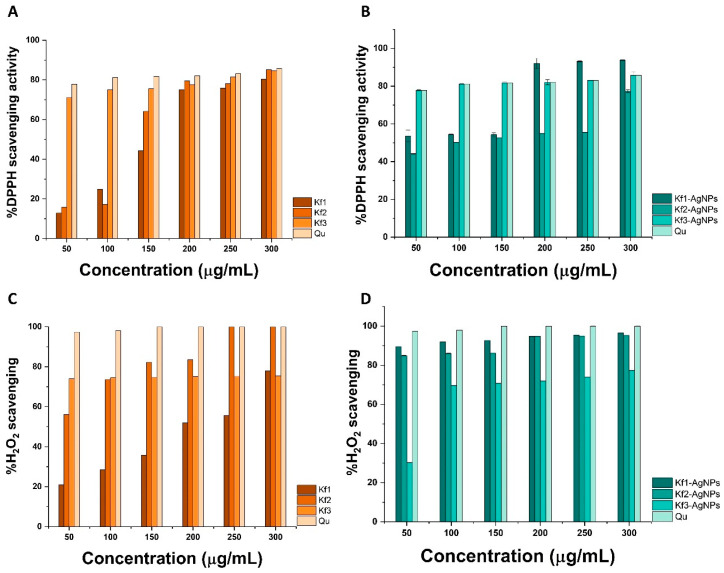
DPPH and H_2_O_2_ scavenging activity of (**A**,**C**) hexane (Kf_1_), chloroform (Kf_2_), and methanol (Kf_3_) extracts from *K. fedtschenkoi*, and (**B**,**D**) synthesized AgNPs with each extract: Kf_1_-, Kf_2_-, and Kf_3_-AgNPs. In both cases, quercetin (Qu) was utilized as the positive control. The mean ± SD of three independent experiments is shown.

**Figure 7 biomolecules-14-00782-f007:**
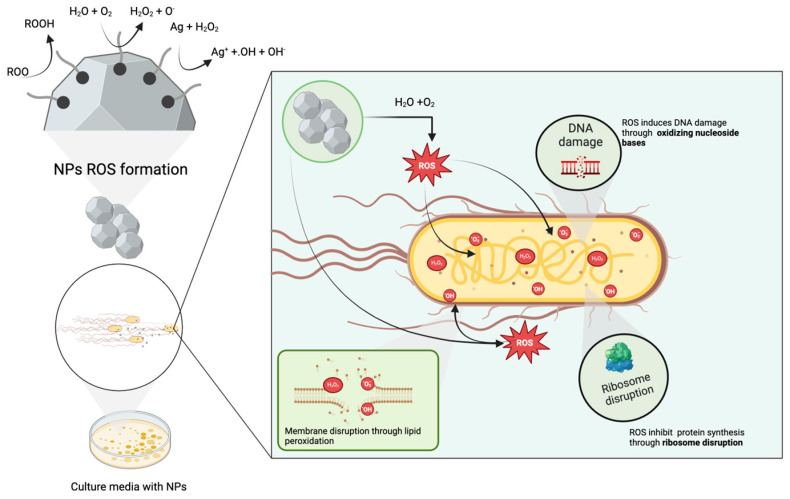
Potential antibacterial and antioxidant mechanisms of green-synthesized AgNPs from *K. fedtchenkoi*.

**Table 1 biomolecules-14-00782-t001:** Calculated MIC_90_ values of extracts and AgNPs from *K. fedtschenkoi* against *E. coli*, *P. aeruginosa*, and *S. aureus*. Concentrations are expressed in μg/mL.

Sample	*E. coli*	*P. aeruginosa*	*S. aureus*
Kf_1_	1149.26	438.41	365.48
Kf_2_	2297.70	425.36	395.24
Kf_3_	1454.30	254.27	1055.16
Kf_1_-AgNPs	1022.63	2282.87	298.16
Kf_2_-AgNPs	844.26	2579.91	421.77
Kf_3_-AgNPs	1219.09	366.21	352.92

**Table 2 biomolecules-14-00782-t002:** Calculated IC_50_ of extracts and AgNPs from *K. fedtschenkoi* in the DPPH and H_2_O_2_ assays. Concentrations are expressed in μg/mL.

Sample	DPPH Assay	H_2_O_2_ Assay
Kf_1_	164.54	197.52
Kf_2_	153.64	1.02
Kf_3_	1.92	3.54
Kf_1_-AgNPs	57.02	3.15
Kf_2_-AgNPs	119.38	2.80
Kf_3_-AgNPs	2.09	3.45

## Data Availability

Data generated in this work can be obtained from the corresponding authors upon reasonable request.

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
