# Peer review of "Green Synthesis of Silver Nanoparticles with Extracts from Kalanchoe fedtschenkoi: Characterization and Bioactivities"

_biomolecules, 2024, doi:10.3390/biom14070782_

Round 1

Reviewer 1 Report

Comments and Suggestions for Authors

Jorge et al prersented a work titled, "Green Synthesis of Silver Nanoparticles with Extracts from Kalanchoe fedtschenkoi: Characterization and Bioactivities". The manuscript describes the green synthesis of silver nanoparticles, characterization and biological evaluation of antibacterial and antioxidant activities. 

 In earlier published paper, author reported the bioactive constituents of Kalanchoe fedtschenkoi and in this manuscript authors described nanoparticle synthesis and biological evaluation. In my opinion, this is a routine work involving metal-nanoparticles conjugated plants extract and bioevaluation except the herbal plant of interest.

The following comments need to be addressed by the authors:-

1. Hexane and chloroform extract are usually non-polar, considered toxic in the case of biological applications, hence, authors need to clarify the residual solvent content in their extract. Most of the published papers described aqueous, and ethanolic extracts. Authors should clarify what advantage may be expected with hexane, chloroform extracts? 

2. The selected positive control drugs: fosfomycin, vancomycin and amikacin, Why authors selected these and need to provide justification of selecting each? and comparing different extracts with different positive control? 

3. In biological screening of antibacterial evaluation, why not MIC determined? Moreover, mostly NPs killed the bacteria at larger doses, these cannot be considered as potent green-NPs.

Reviewer 2 Report

Comments and Suggestions for Authors

Provide more specific background on the use of plant-based extracts and green synthesis of metal nanoparticles as potential solutions to antibiotic resistance and oxidative stress-related disorders.

Discuss the rationale for selecting Kalanchoe fedtschenkoi as the plant source and briefly mention any previous work on the bioactivities of this plant.

Include more details on the preparation and characterization of the plant extracts, such as extraction yields, phytochemical composition, and preliminary screening of biological activities.

Provide information on the source and characteristics of the bacterial strains used in the antibacterial assays.

Expand the discussion to provide more in-depth interpretation of the results, particularly the relationships between the physicochemical properties of the AgNPs and their observed biological activities.

Discuss the potential mechanisms underlying the antibacterial and antioxidant effects of the AgNPs, drawing from the existing literature.

Consider adding a statement on the potential applications or practical implications of the developed AgNPs.

With the successful implementation of the suggested revisions, I believe this manuscript will be suitable for publication. The authors have conducted a well-designed study that contributes valuable insights into the green synthesis and bioactivities of silver nanoparticles derived from Kalanchoe fedtschenkoi extracts. 

Reviewer 3 Report

Comments and Suggestions for Authors

Authors have reported green synthesis of silver nanoparticles from the organic solvent extracts of Kalanchoe fedtschenkoi. Nanoparticle synthesis, characterization using DLS, UV spectroscopy and FTIR has been adequately reported and explained. The antibacterial and antioxidant efficacy of the generated silver nanoparticles have also been reported with sufficient explanations and conclusions. Some of the areas which needs to be considered for revision are as follows

1. Authors have shown the  SEM images (Figure 4) with a relatively large scale bar of 2 microns. Considering the very small size of silver nanoparticles which average around 40 nm (as per DLS) it will be better to show the zoomed nanoparticles with a smaller scale bar (example : 200 nm or 500 nm).

2. EDS analysis (Figure 4D) shows the presence of copper which also has antibacterial effect. This could potentially interfere with the results obtained for the antibacterial assay. Hence, it will be authentic to  show purification of the generated silver nanoparticles from copper. Several methods are available for the same (example: size exclusion chromatography)

Comments on the Quality of English Language

The quality of English is reasonably good and easy to read.

Round 2

Reviewer 3 Report

Comments and Suggestions for Authors

Comments were addressed by the authors